# The Durability of Recycled Fine Aggregate Concrete: A Review

**DOI:** 10.3390/ma15031110

**Published:** 2022-01-31

**Authors:** Changming Bu, Lei Liu, Xinyu Lu, Dongxu Zhu, Yi Sun, Linwen Yu, Yuhui OuYang, Xuemei Cao, Qike Wei

**Affiliations:** 1School of Civil Engineering and Architecture, Chongqing University of Science & Technology, Chongqing 401331, China; buchangming@126.com (C.B.); 2020206111@cqust.edu.cn (L.L.); 2020206077@cqust.edu.cn (X.L.); 2020206029@cqust.edu.cn (D.Z.); 2020206038@cqust.edu.cn (Y.O.); 2020206051@cqust.edu.cn (X.C.); 2Chongqing Key Laboratory of Energy Engineering Mechanics & Disaster Prevention and Mitigation, Chongqing 401331, China; 3College of Materials Science and Engineering, Chongqing University, Chongqing 400044, China; linwen.yu@cqu.edu.cn; 4China Metallurgical Construction Engineering Group Construction Ltd., Chongqing 400084, China; weiqike@cmccltd.com

**Keywords:** recycled fine aggregate concrete, impermeability, drying shrinkage, chloride penetration resistance, carbonation resistance, acid resistance, resistance to freeze–thaw cycles

## Abstract

With the rapid development of urbanization, many new buildings are erected, and old ones are demolished and/or recycled. Thus, the reuse of building materials and improvements in reuse efficiency have become hot research topics. In recent years, scholars around the world have worked on improving recycle aggregates in concrete and broadening the scope of applications of recycled concrete. This paper reviews the findings of research on the effects of recycled fine aggregates (RFAs) on the permeability, drying shrinkage, carbonation, chloride ion penetration, acid resistance, and freeze–thaw resistance of concrete. The results show that the content of old mortar and the quality of recycled concrete are closely related to the durability of prepared RFA concrete. For example, the drying shrinkage value with a 100% RFA replacement rate is twice that of normal concrete, and the depth of carbonation increases by approximately 110%. Moreover, the durability of RFA concrete decreases as the RFA replacement rate and the water–cement ratio improve. Fortunately, the use of zeolite materials such as fly ash, silica fume, and meta kaolin as surface coatings for RFAs or as external admixtures for RFA concrete had a positive effect on durability. Furthermore, the proper mixing methods and/or recycled aggregates with optimized moisture content can further improve the durability of RFA concrete.

## 1. Introduction

With the rapid economic and social development of the 21st century, the urbanization process has produced a large amount of construction waste. According to statistics, China produces more than 3 billion tons of construction waste each year, and waste concrete accounts for 50–60% of all construction waste [1,2]. Such a large amount of waste concrete is usually placed in landfills and roadbed backfills and not effectively utilized, which is not only a waste of resources but also greatly damages the environment and goes against the path of sustainable development [3,4]. Therefore, researchers at home and abroad have conducted prospective research on the application of waste concrete in construction [5,6,7,8,9,10,11,12,13], e.g., making recycled aggregates and geopolymers from waste concrete [14,15,16]. The geopolymer is a new environmentally friendly material that can replace cement in construction. Some people believe that it is the same as, or subordinate to, alkali-activated materials [17]. However, this is wrong, as the two are essentially different processes and produce different chemical products. An alkali-activated material is a monomeric precipitate, whereas a geopolymer is a true polymer [18]. Furthermore, ground polymerization processes require alkaline and acidic environments, whereas alkali-activated materials require mainly high alkalinity environments.

Currently, the mechanical properties and durability of recycled aggregate concrete (RAC) have been studied in depth. A comprehensive review of the mechanical properties of RAC has been conducted, and the results show that degradation of the mechanical properties of RAC is mainly related to the high water absorption of the aggregates and the weak areas formed by the mortar adhering to the old interface [19,20]. The durability of RAC has also been investigated, and the results show that the higher the recycled aggregate (RA) content, the mortar content attached to the RA, and the W/C (water-to-ash ratio), the worse the durability of RAC [21].

Recycled aggregates can be divided into recycled coarse aggregate (RCA) and recycled fine aggregate (RFA). Compared to recycled coarse aggregate concrete (RCAc), recycled fine aggregate concrete (RFAc) has been less studied, but RFA is also important in replacing sand in the preparation of concrete. The application of RFA has a positive impact on the environment [22]. However, the application of RFA is limited in various ways due to its many drawbacks. It was found that the high water absorption of RFA reduces the workability of the prepared fresh concrete. In coarse aggregate concrete, an increase in the water-to-ash ratio (W/C) or the addition of additives or mineral additives is required to offset the lower workability of the RAC [23]. Regarding fresh state characteristics of recycled aggregate concrete, to achieve the same target slump, both the effective and apparent w/c ratios should increase as the amount of RCA incorporated increases. Fine RCA concrete mixes require more water than coarse RCA concrete mixes [24]. The RFA provides more and finer powder material and a more complex interfacial transition zone (ITZ) [25]. Since the old mortar and cement paste attached to RFA severely affect the performance of RFAc, removing them can effectively increase the quality of RFA. A variety of old mortar removal techniques have been studied by domestic and foreign scholars. The mechanical treatment method is simple and easy to operate [26]; however, it tends to crack the RA and has a negative impact on the workability and compressive strength of the concrete. The heat treatment method dehydrates the old mortar adhering to the surface of the RA and renders it loose and porous [26,27]; however, high-temperature heating is highly energy-intensive. The acid presoaking method effectively breaks down the bond between the hydration products of the cement, thus removing the old mortar adhering to the RA surface [28]; however, this method increases the cost of concrete production. In the ultrasonic cleaning method, the cleaned RAC has a 7% increase in compressive strength compared to the uncleaned RAC [29]. Microwave-assisted beneficiation can create higher thermal stresses on the bonding surface of the old mortar and aggregate, thus causing delamination, and this method is less energy-intensive than conventional heating and machine grinding [30]. Some studies have broken away from the tradition and invented the process of crushing all waste concrete to obtain fully recycled fine aggregates [31]. The fully recycled fine aggregate obtained by this process has excellent physical properties and can replace 100% of the river sand in concrete [32].

In this paper, the durability of RFAc is reviewed in terms of impermeability, drying shrinkage, carbonation resistance, chloride penetration resistance, acid resistance, and freeze–thaw resistance. Understanding the effect of RFA on the durability of concrete can provide a theoretical basis for the future large-scale application of RFA as a substitute for sand in practical projects.

## 2. Impermeability

Impermeability is a characteristic of the durability of concrete and is closely related to other durability characteristics. For example, water-soluble chloride ions, CO_2_, and sulfate can enter concrete, or other external factors can cause damage to the concrete, making the service life of concrete greatly affected and thus becoming a global concern [33,34,35]. The impermeability of concrete is mainly determined by its own microscopic properties such as pore size and interfacial transition zone. The impermeability of RFAc is mainly influenced by the replacement rate of RFA, W/C, and mineral admixtures. It was found that the addition of admixtures and additives to concrete can improve the microscopic effect and achieve improved impermeability [36,37,38].

The substitution rate of RFA is closely related to its physical property index. The permeability of the concrete prepared by replacing RFA with different physical properties is also different. Compared with natural fine aggregates, the physical properties of RFA are poor. The higher the substitution rate, the worse the performance of the prepared RFAc. The poor grading of RFA particles with a large number of voids on the surface and micro cracks attached to the old mortar renders the prepared RCAc less dense, so the level of RCAc impermeability decreases with the increase in the RCA substitution rate [39,40]. In self-compacting concrete (SSC), water absorption increases as the RFA substitution rate increases, which is caused by the high porosity and high water absorption of RFA [41,42]. In a sulfate environment, RFAc exhibited better permeability resistance than normal concrete. The water absorption of RFAc exposed to a sulfate environment decreased by an average of 25% compared to normal concrete [43].

The water–cement ratio is an important parameter in the preparation of concrete, which will affect the permeability resistance of concrete. A water–cement ratio of 0.65 has a lower impermeability rating than RFAc with a water–cement ratio of 0.55, indicating that the higher the water–cement ratio, the worse the impermeability of RFAc, because the higher the water–cement ratio, the slower the hydration of the RFAc cement and the more pores it produces, which leads to the poor compactness of RFAc [39,40,44].

Due to its volcanic ash nature, fly ash (FA) has been widely used as a replacement for silicate cement in engineering construction [45]. The addition of fine fly ash to concrete can promote hydration reactions, zeolite reactions, accumulation effects, and nucleation effects [46]. Thus, FA can participate in the early hydration reaction and increase the hydration products to fill the pores and increase the density of the concrete [47]. In self-compacting concrete mixed with 100% RFA (SSC) and 10% (FA), the permeability of SSC prepared with different water–cement ratios was found to be significantly lower compared to the control group without FA [44]. The physical and chemical properties of RFA are defective, which weakens the transition zone at the interface of the prepared RAC. In order to improve the ITZ of RAC, several researchers have developed different concrete mixing methods. Figure 1 shows the double mixing method (DM) [48] and the triple mixing method (TM) [49] as well as the optimized triple mixing method (OTM). The difference between the OTM and the other two methods is that the water-reducing agent (SP) is added in a different order. In the OTM, SP is added together with other gelling materials to promote additional gelling. The dispersion of this material further increases the content of additional gelling material in the ITZ, which can better exploit the zeolite effect at a later stage, such that the use of the OTM can result in a permeability rating of 8 for RAC [50].

The water absorption rate of RFA is much higher than that of natural fine aggregate. RFA absorbs water when freshly mixed with concrete, which has an effect on the mechanical properties and durability of concrete, so the water content of RFA before mixing should not be neglected. Therefore, the effect of RFA in three states of air-drying (AD), oven-drying (OD), and saturated surface drying (SSD) on the permeability of the prepared RFAc was investigated, and it was found that the permeability of these states at the same RFA replacement rate was ordered as follows: SSD > AD > OD. This finding was different from that of RCA. The permeability resistance of concrete prepared from RCA with different moisture contents was ordered as follows: SSD < AD < OD [51]. The reason for this may be that the particle size of RFA is smaller than RCA, but the specific surface area is larger, so the water can become more confined [52].

The addition of RFA has a serious effect on the permeability of concrete. An increase in the RFA replacement rate and the W/C ratio will lead to a decrease in the permeability resistance. However, the use of saturated surface-dried RFA, optimized mixing methods, and the addition of fly ash can improve the impermeability of RFAc. In addition to fly ash, silica fume, metakaolin, and other mineral admixtures also have zeolite properties, so the impermeability of RFAc mixed with silica fume and higher mineral admixtures is also beneficial. Table 1 summarizes the effects of different factors on the impermeability of RFAc.

## 3. Drying Shrinkage

Drying shrinkage requirements are already an important part of project specifications [53]. The evaporation of water from the interior of concrete in a dry environment results in a change in concrete dimensions with time. The drying shrinkage process starts with the evaporation of free water from the pores, followed by the evaporation of adsorbed water from the very small capillary pores; the evaporation of free water is not related to drying shrinkage, but the evaporation of adsorbed water leads to the shrinkage and internal generation of tensile stresses to the extent that drying shrinkage cracks appear in the concrete [54]. As for the coarse aggregate concrete, it was found that the “electric arc furnace slag” improves the strength and modulus of elasticity and leads to shrinkage reduction of concrete [55,56].

The drying shrinkage value increases as the RFA substitution rate increases, and when the substitution rate is 100%, the drying shrinkage value of RFAc is twice as high as that of normal concrete [57]. The drying shrinkage value of RFAc increases with the substitution rate and decreases as the interim period increases. One study found that the drying shrinkage value of RFAc with a 30% substitution rate was 14.0% lower than that of normal concrete and 8.2% higher than that of normal concrete with a 100% substitution rate at 90 days of the prophase [58] (Figure 2). It has also been found that the relationship between drying shrinkage values and the RFA substitution rate is divided by 30%; drying shrinkage values after a substitution rate of 30% increase as the substitution rate increases, and the substitution rate before a substitution rate of 30% has no significant effect on drying shrinkage values [59].

Different sources of RFA also have an important effect on RFAc, and the study of RFAc prepared from different sources of RFA has great significance for the application of waste concrete in practical engineering. RFA from recycled laboratory-produced concrete and RFA from recycled precast concrete, both of which have similar strengths, were studied for drying shrinkage values of RFAc under both recycling routes and were found to be similar [60]. However, in another study, which also investigated RFA from two different sources, RFA1 from flat-plate concrete and RFA2 from composite beam specimen concrete, drying shrinkage tests on RFAc prepared from both RFAs at 0%, 50, and 100% substitution rates were conducted, and it was found that, regardless of the substitution rate of RFA, the drying shrinkage values of RFAc prepared from RFA2 were lower than that of RFAc prepared by RFA1. The differences in drying shrinkage values exhibited by RFAc prepared by the two sources are mainly related to the physical properties of RFA: (1) the density of RFA2 is greater than that of RFA1 (RFA1: 2222 kg/m^3^, RFA2: 2326 kg/m^3^), indicating that RFA1 contains a higher amount of old cement paste than RFA2; and (2) the water absorption is lower than that of RFA1 (RFA1: 11.2%, RFA2: 8.1%). These two differences in physical properties indicate that the water content of RFAc prepared by RFA1 is higher than that of RFAc prepared by RFA2 and that the high water content has a low binding effect on concrete [61] (Figure 3).

The drying shrinkage of RFAc with a 20% RFA replacement rate and an effective water–cement ratio of 0.43 was similar to that of the control with a water–cement ratio of 0.45 at the same amount of RFAc mixing water, but the drying shrinkage of RFAc with a 30% RFA replacement rate and an effective water–cement ratio of 0.41 was lower than that of the control [62]. SP is a very frequently used admixture, and its addition reduces the amount of mixed water in the workable fresh concrete mixture, reducing the water–cement ratio for the same amount of cement and thus improving the performance of RFA [63,64,65]. The effect of different SPs on the drying shrinkage of RFAc prepared with different replacement rates of RFA was investigated, mainly comparing RFAc without SP, with conventional plasticizer (SP1), and with high-performance and high-efficiency plasticizer (SP2), and it was found that the drying shrinkage values of RFAc without SP increased to 28 and 57% at 7 and 91 days, respectively; the drying shrinkage values of RFAc with SP1 increased at 7 days to 44%, while the drying shrinkage value at 91 days decreased to 2%; the drying shrinkage value of RFAc7d with SP2 increased to 38%, while the drying shrinkage value at 91 days decreased to 30%, which shows that the addition of SP was effective in reducing the drying shrinkage value of RFAc, and the effectiveness of SP2 was higher than that of SP1 [66].

Silica fume (SF), produced during the smelting process of the ferrosilicon industry, has a silica content of up to 75% and is an effective volcanic ash material [67]. Due to the large amount of water absorption of the old mortar attached to the RA surface, it can lead to an increase in the drying shrinkage value of SSC prepared with RFA. It was found that the drying shrinkage value of the ternary RFA SSC prepared with finely ground blast furnace slag (GGBFS) and SF instead of cement was greatly reduced, on the one hand because the two external admixtures promoted hydration and on the other hand because the ultrafine material filled the pores inside the concrete [44].

Different mixing methods also have an effect on the performance of concrete. RFA is one of the important reasons for the performance of concrete due to its excessive water absorption; however, the use of moisture in the two-part addition, i.e., the two-stage mixing method (TSMA), can effectively prevent the excessive water absorption of RFA [68]. The performance of concrete with different RFA replacement rates prepared by the normal mixing method and TSMA was investigated, and it was found that the drying shrinkage values of all RFAc using TSMA were lower than those of the concrete prepared by the normal mixing method, mainly because of the higher compactness and better ITZ of RFAc prepared by TSMA [69].

In summary, the drying shrinkage values of the concrete prepared by the RFA substitution of sand became larger with the increase of the RFA substitution rate. In order to reduce the effect of RFA on the drying shrinkage value of concrete, the drying shrinkage value can be effectively reduced by adding admixture (SP) and mineral admixture, and the drying shrinkage value of RFAc with TSMA is also smaller than that of RFAc with a normal mixture. Drying shrinkage is a long-term performance feature of concrete, which is of great significance to concrete in the process of use, but RFAc is significantly affected by RFA, so it is necessary to consider an optimal replacement rate of RFA when preparing RFAc. Therefore, an optimal replacement rate of RFA needs to be considered when preparing RAFc, and the effect of RFA on the drying shrinkage of concrete should be reduced by other means.

## 4. Carbonation Resistance

The amount of CO_2_ in the air is about 0.03%. The pores present in concrete allow CO_2_ to diffuse within it and react with hydration products to form a carbonation process. Carbonation is mainly a chemical reaction between carbon dioxide (CO_2_) and calcium hydroxide (CH), hydrated calcium silicate (C-S-H), or bentonite (AFt) [70]. Determining the carbonation resistance of RFAc is fundamental to the study of its durability, since concrete is exposed to natural environments that induce carbonation corrosion for long periods of time, and the rate of carbonation is directly related to the lifetime of concrete. The carbonation resistance of RFAc is mainly related to the minimum particle size of RFA, the substitution rate, the W/C ratio, and the mineral external admixtures.

The smaller the RFA particle size, the higher the amount of old mortar adhered to the surface, so the carbonation depth of RFAc increases instead as the minimum particle size of the RFA decreases [71]. CO_2_ curing is one of the means of treating RFA, and CO_2_ curing can fill the pores and cracks on the surface of the aggregate and improve the performance of the old and new ITZ [72]. Carbonation studies were conducted on concrete prepared from CO_2_-cured recycled coarse and fine aggregates; it was found that the treated RFAc had improved carbonation resistance compared to the untreated one, with the maximum carbonation resistance depth of the former being about 42.5% lower than that of the latter (Figure 4), and it also showed the same trend under freeze–thaw cycling conditions [73].

When the replacement rate of RFA in concrete is 30 and 100%, the carbonation depth of concrete CO_2_ increases by about 40 and 110%, respectively, indicating that the resistance of RFAc to carbonation decreases as the replacement rate of RFA increases, which is caused by the internal capillarity of RFAc [71,74,75]. In self-compacting concrete (SSC), the depth of carbonation also increases with the increase in the substitution rate [76].

The lower carbonation resistance of RFAc compared to the control is not only due to the high porosity of RFA but also related to the increase in the effective W/C, which improves the diffusion of CO_2_ [77]. Fly ash is beneficial for the anticarbonation properties of RFAc; when fly ash replaced cement in the range of 10–30%, a replacement rate of 20% was found to be most beneficial for the anticarbonation properties of RFAc [71]. Some studies, however, have found that the carbonation depth of RFAc increased with increasing fly ash when the RFA substitution rate was below 30% [78]. In self-compacting concrete (SSC) incorporating RFA, it was found that partial age soils (MK) were beneficial to the anticarbonation properties, and SSC with a 50% RFA substitution rate and incorporating MK showed a 20 and 18% decrease in carbonation depth after 4 and 12 weeks of accelerated carbonation exposure, respectively, while SSC with a 100% RFA substitution rate and incorporating MK showed the same carbonation depth after 4 and 12 weeks of accelerated carbonation exposure as the control group. The reason for this may be the high reactivity of the added MK and volcanic ash products such as calcium hydroxide, which caused a large consumption of early hydrated calcium (CH) and thereby reduced the carbonation depth [76].

The carbonation depth increased with time, and the RFAc with SP2 (SikaPlast898) addition had a lower carbonation depth than both without plasticizer and with SP1 (Sikament 400 plus) RFAc at each prophase, mainly because the addition of SP2 decreased the effective w/b of RFAc. Thus, the internal porosity was lower, and the gas permeability was lower [79]. The resistance to carbonation of recycled concrete prepared by one generation of RFA (RAC1) and recycled concrete prepared by two generations of RFA (RAC2) was investigated at four bending stress levels (40, 70, 100, and 120%), and it was found that the carbonation depth increased with stress, except for the 70% stress level, and the carbonation depth of RAC2 was higher than that of RAC1 [75].

The particle size of RFA affects the carbonation resistance of RFAc, and the increase of the RFA substitution rate and W/C can negatively affect the carbonation resistance of RFAc. The anticarbonation performance of RFAc can be improved by the CO_2_ curing of RFA and the addition of mineral ex-dopant. CO_2_ curing is one of the methods that can be used to improve the quality of RFA, and mineral ex-dopant coating and microbial carbonate precipitation treatment can also improve the quality of RFA and thus the anticarbonation performance of RFAc. Table 2 summarizes the influence of different factors on the anti-carbonation performance of RFAc.

## 5. Chloride Penetration Resistance

Buildings in harsh marine environments are subjected to long-term chloride ion corrosion, leading to the degradation of reinforced concrete structures, which can seriously affect the service life of buildings [80,81]. There are two main effects of chloride ions on buildings: (1) the physical salt attack of chloride ions causes cracking and scaling on building surfaces [82]; and (2) chloride ions corrode the reinforcement inside the building, penetrate the protective layer of concrete, reach the reinforcement, and gradually accumulate, causing the demineralization of the reinforcement, which corrodes the prebuilt reinforcement [83]. RFA has various poorer physical properties compared to natural sand, so the chloride ion permeability resistance of RFAc is more importantly affected by the RFA substitution rate, curing conditions, CO_2_ curing, and mineral admixtures.

The chloride ion diffusion coefficient and electric flux of RFAc are much higher than those of control groups and show a linear growth trend with the increase of the RFA substitution rate. Reasons include the following: (1) a large amount of old mortar attached to the surface of RFA leads to high porosity, which provides a channel for chloride ion transmission; (2) the old mortar on the surface of RFA absorbs water and reaches the saturation surface, which prevents the transition area of the old and new mortar interfaces from becoming dense, causing it to form chloride ion transmission channels; and (3) additional water cannot be completely absorbed by RFA, resulting in a water–cement ratio that is higher than the effective water–cement ratio and thus a low compactness in the RFAc [40,84]. The total charge of SSC increased significantly with an increasing RFA substitution rate, but the resistivity and chloride resistance of SSC at a 25% substitution rate did not change significantly compared to a control [41]. The RAC prepared by considering coarse and fine recycled concrete aggregates in plain concrete, because of the high porosity of the aggregate itself, was accompanied by a high diffusion coefficient and increased conductivity, so the conductivity of the prepared RAC increased with increases in the substitution rate of the coarse and fine recycled concrete aggregates [85,86].

Curing is an important stage in the early stages of concrete. Reactive powder concrete (RPC) was cured using two curing conditions, normal curing and thermal curing, and the resistivity of RPC was found to be higher under thermal curing conditions, which is due to the improved volcanic ash reaction of RPC under thermal curing conditions, producing a denser CHS gel that fills the pores [87].

The properties of RFAs obtained by different treatments are different, and the prepared RFAc exhibits different properties. The resistivity of RFAc was studied using two types of RFA, (1) recycled fine aggregate and recycled coarse aggregate mixed together (R1) and (2) fully recycled fine aggregate (R2), and it was found that the resistivity of RFAc mixed with R2 was higher than that mixed with R1 due to the improved physical properties and the lower water absorption of R2 [86]. CO_2_-cured RFA and untreated RFA were compared to study the effect of both on the chloride ion permeability of RFAc, and it was found that the chloride ion permeability of RFAc prepared from CO_2_-cured RFA was higher at all substitution rates (Figure 5), and the reason for the reduced chloride ion resistance of CO_2_-cured RFAc was as follows: When RFA was carbonated, chloride ions were attached as Friedl′s salt on the cement paste and then released into the pore solution, leading to an increase in chloride ion concentration. As carbonation proceeded, the chloride ion concentration in the carbonated zone increased, while the chloride ion concentration gradient in the noncarbonated zone also increased, so chloride ions entered the noncarbonated zone, leading to an increase in chloride ion concentration in the noncarbonated zone [88].

The chloride ion resistance of the concrete added to RFA was generally reduced, and it was found that the chloride ion resistance of RFAc could be improved by adding an external admixture. The addition of 10% silica fume (SF) to self-compacting RFA concrete had a lower chloride ion permeability than the control group for all prophases, and this was because SF acts as a microfiller, increasing the density of the SSC interface transition zone [89]. A high amount of clay produces a volcanic ash material called metakaolin (MK) when calcined, which is used as an additional cementitious material (SCM) in concrete because of its volcanic ash nature. The addition of MK to mortar and concrete has been found to improve mechanical and durability properties [90,91]. The chloride ion permeability of regenerated SSC with the addition of 10% MK decreased in all properiods, where C100F25M with a 25% minimum RFA substitution decreased by 64% relative to the control group for 28 days, and C100F100M with a 100% maximum RFA substitution decreased by 52% relative to the control group for 28 days, and this was because of the volcanic ash effect of MK and the additional hydration products, which resulted in a denser microstructure of the regenerated SSC [42]. Rice husk ash (RHA), obtained after combustion treatment, can be about 90% SiO_2_ and has high volcanic ash properties [92]. RHA is used as a cement replacement for concrete because of its high SiO_2_ content and volcanic ash effect [93,94,95]. It has been found that RHA can reduce chloride ion penetration in concrete [20]. The durability of RFAc was greatly affected by the addition of recycled aggregates, in which the resistance to chloride ion penetration was significantly reduced. Therefore, finely ground rice husk ash (GRHA) was used as a cementitious material to prepare RAC with a 20–50% replacement of cement, and it was found that the coarse and fine aggregates were completely replaced by RA in RAC at 90 and 180 days. The depth of chloride ion penetration decreased with the increase of GRHA admixture, as shown in Figure 6. This was mainly because the CSH generated by the reaction between GRHA and volcanic ash filled the pores and because the SiO_2_ and CSH in GRHA impeded the movement of chloride ions [96]. The durability performance of high-performance recycled aggregate concrete (HPRAC) prepared with ground blast furnace slag (GGBFS) and recycled aggregates is also of interest, so the effect of fly ash (FA) and MgO on the durability performance of HPRAC was investigated, with an RCA replacement rate of 40%, with 30% RFA in HPRAC, with 0, 15, 30, and 45% FA dosing to replace GGBFS, and at 15% FA doping, with 2.5, 5, and 7.5% MgO. It was found that the chloride permeability of HPRAC was reduced with the increase of doping after 15% FA doping. This is because the appropriate amount of FA substitution for GGBFS can fill the pores to promote hydration reactions, but when the FA doping is too high, FA exhibits particle stability but will reduce the hydration rate; in contrast, GGBFS has the best impermeability performance with the addition of 5% MgO chloride ions in a 15% FA admixture, and this is because MgO promotes the AAS hydration reaction, which fills the pores of concrete and thus increases the density of the structure [43,97].

Surface coating treatment is one of the methods used to enhance the performance of RFA. The RFA was coated with fly ash, slag, and PVA (polyvinyl alcohol), and 25% of the natural sand was replaced with the coated RFA. It was found that the resistivity of the slag-coated RFA was maximal, regardless of the w/b, and increased with the thickness of the coating [25].

The addition of SP can improve the durability performance of concrete, and the addition of SP in RFAc can increase the density of the concrete, so the durability performance of RFAc can be improved. However, it was found that there was no significant difference in chloride ion penetration between RFAc with SP and those without [98].

The resistance of RFAc to chloride ion permeation was decreased with the increase of the RFA substitution rate. The treatment of RFA with CO_2_ was detrimental to the chloride ion permeation resistance of RFAc, but the coating-treated RFA was positive against chloride ion permeation, and the addition of mineral admixtures enhanced the chloride ion permeation resistance of RFAc. It is impossible to avoid the reduction of the chloride ion permeation resistance of concrete by adding RFA, so we should consider upgrading the quality of RFA itself or adding mineral admixtures and SP to mitigate the reduction in its performance. Table 3 summarizes the effects of different factors on the resistance of RFAc to chloride ion penetration.

## 6. Acid Resistance

Industrial production causes a large amount of sulfur dioxide emissions, as well as an increasing number of nitrogen oxides from vehicle exhaust, which are the main components that contribute to the production of acid rain (NO_x_) and sulfur dioxide (SO_2_) [99]. The concrete structure of buildings exposed to acid rain for long periods of time is affected by the deterioration of concrete and the rusting of reinforcements. The deterioration of concrete by acid is the reaction of sulfuric acid with calcium hydroxide to produce gypsum, as shown in Equations (1)–(3), and gypsum formation expands concrete and accelerates corrosion. RFAc containing 100% RFA is less durable than natural sand concrete because of such properties as the internal capillarity of RFAc [100].
Ca(OH)_2_ + H_2_SO_4_ → CaSO_4_ · 2H_2_O(1)
CaO · Al_2_O_3_ · 12H_2_O + 3(CaSO_4_ · 2H_2_O) + 14H_2_O → CaO · Al_2_O_3_ · 3CaSO_4_ · 32H_2_O(2)
CaO · SiO_2_ · 2H_2_O + H_2_SO_4_ → CaSO_4_ + Si(OH)_4_ + H_2_O(3)

The acid resistance of concrete depends mainly on its percentage reduction in mass and compressive strength in acid solutions. The reduction in mass loss and compressive strength in a 3% sulfuric acid solution was discussed by adding RFA with different substitution rates to the activated concrete, which was considered from two sources, one from normal concrete and the other from activated powder concrete. It was found that the mass loss and compressive strength of the activated powder concrete prepared from both sources of RFA subjected to sulfuric acid attack increased as the substitution rate of RFA increased, but both had good acid resistance [87]. This is mainly due to the reaction of sulfuric acid with calcium hydroxide through the porous activated powder concrete to produce gypsum, which expands calcium salts in the pores in aqueous solutions and finally leads to a decrease in the mass and compressive strength of the activated powder concrete [101]. The acid resistance of high-performance concrete (HPC) in sulfate solutions and sulfuric acid solutions was studied by adding 20% RCA and RFA, and it was found that HPC had good acid resistance in sulfate solutions, probably due to the addition of SF, but a severe loss of strength in sulfuric acid solutions [102]. In concentrated hydrochloric acid and concentrated sulfuric acid, the corresponding mass loss of plain concrete with added RFA increased with the soaking time and the substitution rate, and the mass loss of RFAc with a 30% substitution rate after 90 days was lower than that of the control, while the mass loss of RFAc with a 100% substitution rate was higher than that of the control. This was because the RFA with a 100% substitution rate was finer and contained more pores and old cement mortar [58]. Similar findings were found in another study on the effect of RFAs on the acid resistance of alkali-activated slag concrete (AASC), a new environmentally friendly cementitious material, where AASC with 0, 50, and 100% RFA replacement rates were exposed to a 3% sulfuric acid solution for 180 days, and it was found that the quality of AASC decreased as the replacement rate increased, but even after 180 days of exposure, the maximum mass loss of AASC was less than 20% (as shown in the figure), which was mainly due to the presence of GGBS in AASC [103] (Figure 7). The acid resistance of SSC prepared with coarse and fine recycled aggregates in 5% sulfate was better than the control. Meanwhile, the effect of natural volcanic ash (PZ) on the recycled SSC was investigated, and it was found that the addition of PZ improved the acid resistance of the recycled SSC, with 20% PZ dosing being optimal [104].

Different types of concrete supplemented with RFA showed mass loss and compressive strength reduction in acid solutions. The more significant the mass loss and compressive strength reduction was, the more the replacement rate of RFA increased. The incorporation of natural volcanic ash into SSC incorporated with RFA improves the acid resistance. An appropriate RFA substitution rate should be considered for RFAc in acidic environments, and a certain amount of natural volcanic ash can be considered to improve the acid resistance.

## 7. Resistance to Freeze–Thaw Cycles

In the UK, freeze–thaw cycles have been identified as one of the most significant deleterious factors affecting the life of concrete structures [11,105,106]. The water inside the concrete under the effect of freeze–thaw cycles turns into micro-ice bodies, with its volume expanding by up to 9% [107]. When there is not enough space for volume expansion in concrete, hydraulic and tensile stresses are generated in the pores, which leads to further increases in pore size [108]. The increased pore space, in turn, draws in more water, which causes greater tensile stresses when it freezes again, eventually leading to concrete deterioration. This shows that pore space and water are closely related to freeze–thaw resistance. The freeze–thaw resistance of RAC is considered to be poor due to the presence of old mortar and micro cracks adhering to the RA surface, resulting in a high water absorption rate [21].

The freeze–thaw resistance of RFAc was investigated by considering three groups of RFAc: plain RFAc with a water–cement ratio of 0.53 (C); high-strength concrete (HC) without SP and with an aerator (AEA), with a water–cement ratio of 0.35; and high-strength concrete (HCAE) with SP added but without an AEA with a water–cement ratio of 0.35, at replacement rates of 0, 20, 50, and 100%. Three hundred freeze–thaw cycles were tested for the three groups of RFAc, and it was found that the mass loss of C was much higher than that of HC and HCAE (Figure 8). This result indicates that the freeze–thaw resistance of RFAc is not related to the type of aggregate but is determined by the water–cement ratio, and the addition of an AEA has little effect on the freeze–thaw resistance of RFAc, as the water–cement ratios of HC and HCAEA were the same but had similar mass loss. Therefore, reducing the water–ash ratio is an effective way to improve the freeze–thaw resistance [109]. In another study, it was found that the mass loss values of RFAc in under 300 freeze–thaw cycles decreased as the water–cement ratio decreased from 0.7 to 0.5 [110].

Internal curing is the internal water supply to concrete to achieve maintenance. Internal curing can reduce water consumption, improve the homogeneity of concrete, and reduce the cracking of concrete surfaces. Studies dedicated to internal curing found that internal curing may have an effect on the durability of concrete structures [111]. RFA has a much higher water absorption than natural aggregates and can be used as a source of internal curing. Applying RFA to the internal curing of concrete was found to be beneficial for properties such as cracking delay, freeze–thaw resistance, and the dynamic modulus of elasticity, and using 50% RFA was recommended for internal curing because the best results were achieved at that substitution rate [112]. The different moisture concentrations of RFA may also have an effect on internal curing, so RFA prepared at 0, 50, and 100% saturation were investigated for RFAc. The performance was investigated, and it was found that the best freeze–thaw resistance was achieved when RFAc was prepared at 50 and 100% saturation for internal conservation [110].

Different substitution rates of RFA also have an important effect on the freeze–thaw resistance of RFAc. By studying the performance of SSC with added RFA, it was found that the instability and the weak ITZ of SSC increased with the increase of the RFA substitution rate, and the compressive strength of 30% of the SSC specimens with added RFA after 28 and 56 days of freeze–thaw cycles decreased to less than 350 MPa [113].

In summary, the freeze–thaw resistance of RFAc is significantly affected by the substitution rate of RFA, but reducing the water–cement ratio can enhance that resistance. In addition, using RFA as a source for internal maintenance and using RFA with a suitable moisture content both have positive effects on this freeze–thaw resistance. RFA itself suffers from a high water absorption, but using it as a source for internal maintenance can be reasonably applied to enhance the resistance of RFAc to freeze–thaw cycles. Table 4 shows the influence of different factors on the freeze–thaw resistance of RFAc.

## 8. Conclusions

This paper focuses on the durability of RFAc. The efficiency of RFA with appropriate water content, pretreated RFA, concrete mixing methods, and mineral admixtures on the durability of RFAc is summarized. Some important points are highlighted here:(1)The durability of RFAc is closely related to the quality of the RFA, which is mainly influenced by the content of the old mortar, specifically, the higher the content of the old mortar, the higher the porosity and water absorption. The average water absorption of natural sand is about 1.1%, while the average water absorption of RFA can reach about 8.4%.(2)The impermeability, drying shrinkage, resistance to chloride ion penetration, carbonation, acid resistance, and freeze–thaw cycle resistance of RFAc decrease as the RFA replacement rate improves. For example, the drying shrinkage value at a 100% RFA replacement rate is twice that of normal concrete, and the depth of carbonation increases by approximately 110%.(3)The impermeability, carbonation resistance, and freeze–thaw cycle resistance of RFAc decrease as the effective water–cement ratio increases. For example, when the water-to-ash ratio is reduced from 0.7 to 0.5, the mass loss of RFAc under freeze–thaw cycles is reduced. Meanwhile, the drying shrinkage is less affected by the change in the water–cement ratio.(4)RFA with appropriate moisture benefits the permeation resistance and freeze–thaw cycle resistance of RFAc.(5)The treatment of RFA with CO_2_ has a positive effect on the anticarbonation performance of RFAc. However, it is unfavorable to the resistance to chloride ion penetration.(6)Advanced mixing methods focused on improving the ITZ have been reported, and they can partially compensate for the poor performance of recycled concrete fine aggregates. Thus, the relative durability of RFAc can be optimized.(7)Existing studies report an optimized triple mixing method, where the SP is added together with additional gelling material. This addition promotes the dispersion of that gelling material, which, in turn, leads to a higher content of additional gel material in the ITZ, a better volcanic ash effect at a later stage, and a reduction in the porosity of the RFAc, thus allowing for an impermeability rating of 8 for the RAC using this method. In addition, because a water-reducing agent is added early on, together with the additional gelling material in this method, a thin layer of additional gelling material can be formed, even when the mixture is insufficient.

## 9. Outlook

The replacement rate of RFA has a serious impact on the durability of concrete. Therefore, when the RFA replacement rate is 100%, 10% FA can be added to improve the impermeability. In terms of drying shrinkage values performs, the best RFA replacement rate of the concrete is 30%; as for how the resistance to chloride penetration performs, the best RFA replacement rate is 25%. The durability of RFAc is closely related to the quality of RFA. In addition to the methods mentioned in this paper to improve the quality of RFA, there are also thermal–chemical, thermal–physical, and microbial precipitation methods; thus, the durability of concrete prepared from RFA by different treatment methods is also worth studying in future research. In addition, other work will be focused on the properties of acid resistance and freeze–thaw cycle resistance. The performance of concrete in harsh environments should also be considered.

## Figures and Tables

**Figure 1 materials-15-01110-f001:**
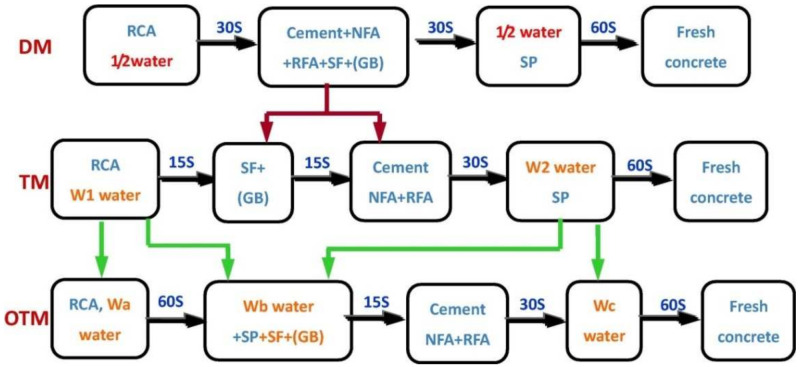
Comparison of mixing methods for the preparation of RAC [50] (Wa equals 60–80% of the product of the weight of RA multiplied by its water absorption, Wb equals the weight percentage of residual water of SCM in the total gelling material, and Wc = mixed water − Wb − Wa).

**Figure 2 materials-15-01110-f002:**
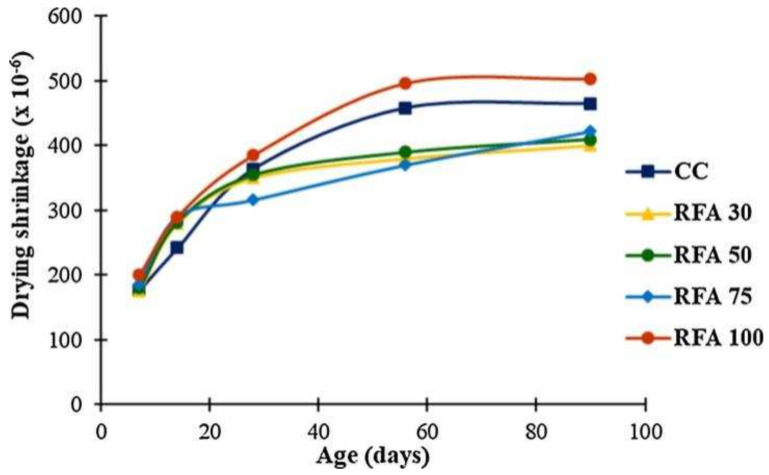
Dry shrinkage of RS (river sand) and RFA concrete [58].

**Figure 3 materials-15-01110-f003:**
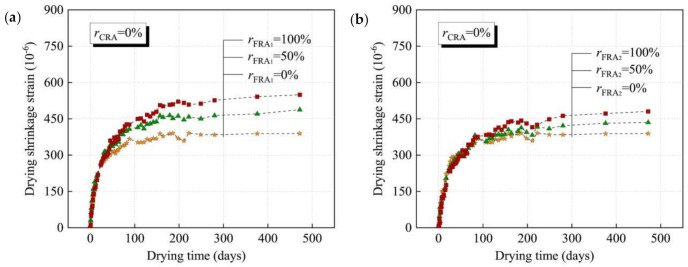
Development of dry shrinkage of concrete with time using different sources of FRA [61]: (**a**) concrete with FRA1, rCRA = 0%; (**b**) concrete with FRA2, rCRA = 0%.

**Figure 4 materials-15-01110-f004:**
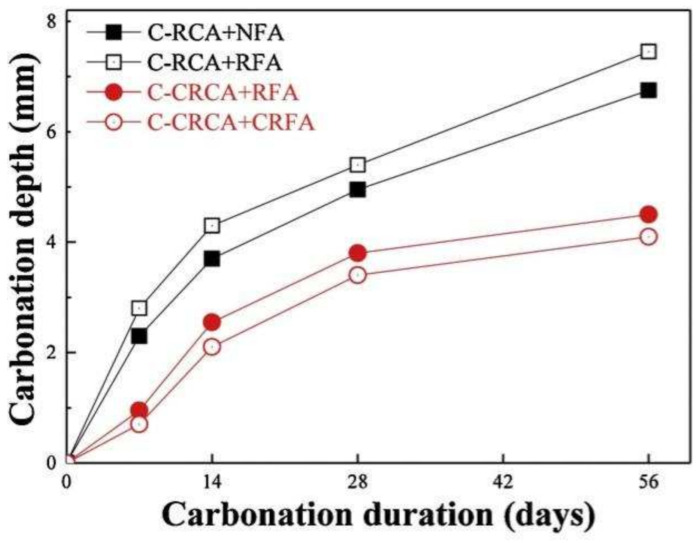
Carbonation behavior of CRA concrete exposed to carbonation [73].

**Figure 5 materials-15-01110-f005:**
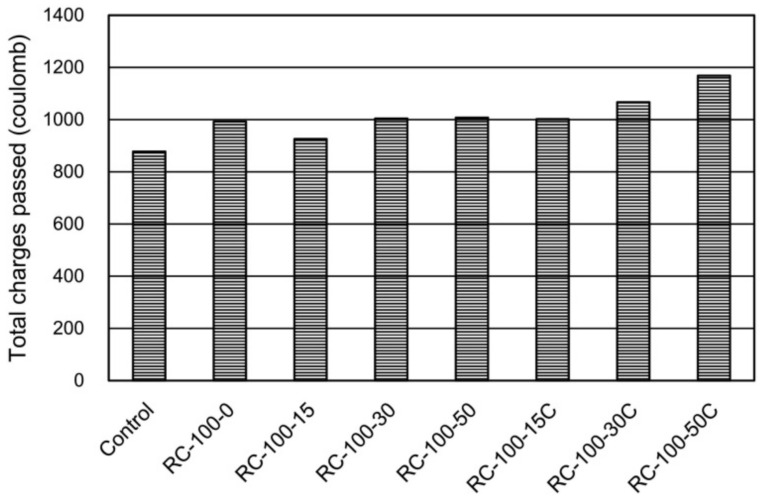
Chloride ion penetration at 56 days [88].

**Figure 6 materials-15-01110-f006:**
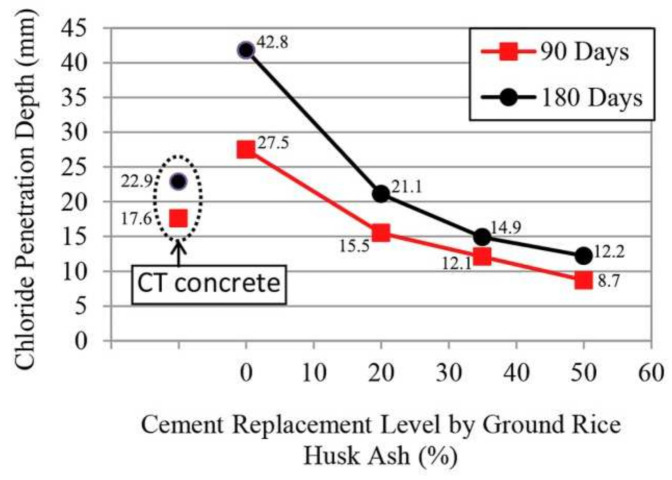
Relationship between the depth of chloride ion penetration in concrete and the level of rice husk ash replacement [96].

**Figure 7 materials-15-01110-f007:**
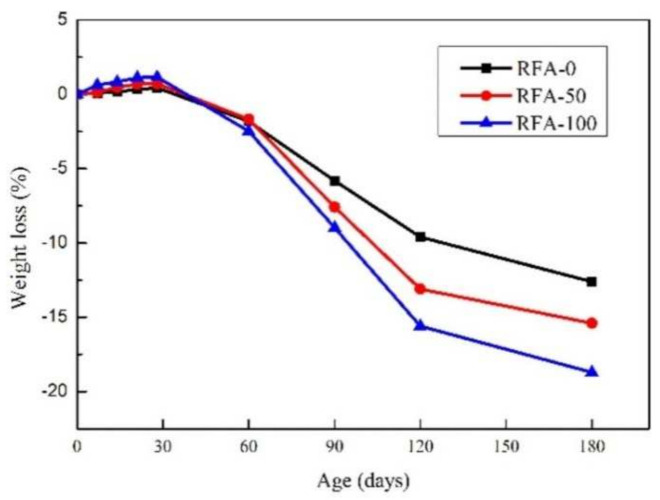
Weight loss of alkali−activated slag concrete [103].

**Figure 8 materials-15-01110-f008:**
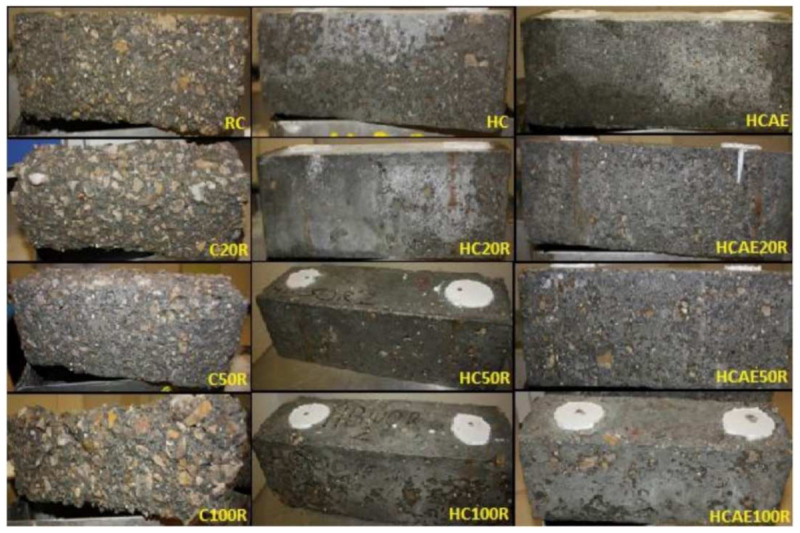
Macroscopic degradation of mixtures after 300 freeze–thaw cycles at different RFA replacement rates [109] (C: plain concrete with a water–cement ratio of 0.53; HC: high-strength concrete without SP and with an aerator (AEA), with a water–cement ratio of 0.35; HCAE: high-strength concrete with SP added but without an AEA, with a water–cement ratio of 0.35).

**Table 1 materials-15-01110-t001:** The influence of different factors on the impermeability of RFAc.

Factor	Change	Influence
RFA %	Increase	Negative
W/C ratio	Increase	Negative
RFA water content	Increase	Positive
Optimized triple mixing method (OTM)	-	Positive
Mineral admixture	-	Positive

**Table 2 materials-15-01110-t002:** The influence of different factors on the anti-carbonation performance of RFAc.

Factor	Change	Influence
RFA minimum particle size	Decrease	Negative
RFA%	Increase	Negative
W/C ratio	Increase	Negative
CO_2_ pretreatment RFA	-	Positive
Mineral admixture	-	Positive

**Table 3 materials-15-01110-t003:** The influence of different factors on the resistance of RFAc to chloride ion penetration.

Factor	Change	Influence
RFA%	Increase	Negative
Thermal curing	-	Negative
CO_2_ pretreatment RFA	-	Negative
Surface coating treatment RFA	-	Positive
Mineral admixture	-	Positive

**Table 4 materials-15-01110-t004:** The influence of different factors on the freeze–thaw resistance of RFAc.

Factor	Change	Influence
RFA%	Increase	Negative
W/C ratio	Decrease	Positive
RFA as an internal conservation source	-	Positive
RFA moisture content: 50%/100%	-	Positive

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
