# Peer review of "The Durability of Recycled Fine Aggregate Concrete: A Review"

_materials, 2022, doi:10.3390/ma15031110_

Round 1

Reviewer 1 Report

The manuscript entitled "Durability of recycled fine aggregate concrete: A review" presents a comprehensive review in the field of geopolymers. Therefore, the study includes multiple valuable information and base concepts related to geopolymers manufacturing and characterization. However, multiple affirmations aren’t supported by the provided references and many other issues must be addressed. The paper needs minor revisions before it is processed further, some comments follow:

A clear statement should be introduced into the abstract and an introduction to highlight with fine aggregates have been considered in this review (as there are multiple aluminosilicate sources for geopolymers, including mine tailings).

In terms of geopolymers, there are disputes about the difference between geopolymer and alkaline activated materials. In some sentences, those are considered the same, while by some authors alkali-activated and geopolymers are considered distinct materials. Authors need to add one or two sentences about this to tell readers that in this article, alkali-activated materials are regarded the same as geopolymers or different. Please introduce corresponding clarifications.

The introduction section must be significantly improved. Multiple affirmations aren’t supported by the provided references. Please introduce the corresponding citation to support the affirmations from the introduction section. The affirmations from the first paragraph have no background in the experimental results obtained (as there are none), therefore, please introduce corresponding citations.

Please carefully check the correlation between the reference and the content of the corresponding publication.

Moreover, the literature is OLD, only three studies published in 2021 have been considered (this is way too low). Please conduct a literature survey of the most recent published studies and consider those as the newest discoveries in the field.

Section 7: Figure 11 – please increase the scientific degree of the figure. Please introduce figure labels to highlight the area of interest for the readers.

 Please introduce a new section to highlight the "future research directions" in the field considering the conclusions drawn in this study. Those highlights will be very important for the researchers from this field.

There are multiple typing errors in the text, please check it carefully.

Reviewer 2 Report

  • The article still needs several grammatical and syntax improvements. Use of English service center is recommended. The language should be improved.
  • The type should be clearly
  • The scope of work and novelty should be clearly mentioned within the abstract.
  • The abstract is written qualitatively. Majority of the qualitative statements should be modified for quantified result comparisons.
  • The authors mentioned about “Such a large amount of waste concrete is usually used in landfills and roadbed backfills without being effectively utilized, which is not only a waste of resources but also greatly damages the environment and goes against the path of sustainable development. Therefore, researchers at home and abroad have done a lot of prospective research on the application of waste concrete in construction”. The following references should be added for this statement comprehensiveness: 1) Temperature and humidity effects on behavior of grouts. Advances in concrete construction. 2) Nano silica and metakaolin effects on the behavior of concrete containing rubber crumbs. CivilEng. 3) Investigation of steel fiber effects on concrete abrasion resistance, Advances in concrete construction.
  • The introduction needs to be revised for higher quality language. The authors mentioned some works without stating about the contributions, pros and cons and the how the current work would address.
  • The authors mentioned” freeze-thaw cycles have been identified as one of the most significant deleterious factors affecting the life of concrete structures” The following references should be added for this statement: 1) Compressive behavior of concrete under environmental effects. IntechOpen. 2) Temperature and humidity effects on behavior of grouts. Advances in concrete construction. Figures from other resources should be appropriately referenced.
  • Legends for Figure 5 should be modified.
  • Each subsection should be generally explained and referenced. Subsequently, the major results for RCA should be provided.
  • The article should have significantly more information and explanation
  • Conclusion should be quantified.
    • The quality of RFA is mainly effected by the content of old mortar. In details, both
      of the higher the content of old mortar, and the higher the porosity and water absorption
      of RFA would worsen the durability of the prepared RFAc.
    • The impermeability, drying shrinkage, resistance to chloride ion penetration, carbonization, acid resistance and freeze-thaw cycle resistance of RFAc would become lower
      with the improvement of RFA replacement rate
    • The impermeability, carbonization resistance and freeze-thaw cycle resistance of
      RFAc decrease with the increase of the effective water-cement ratio. Meanwhile, the drying shrinkage is less effected by the change of the water-cement ratio
  • The optimization evaluation should be described in details

  • Advanced mixing methods focused on the improvement of ITZ have been reported, which can partially compensate the poor performance of recycled concrete fine
    Thus, the relative durability of RFAc would be optimized.
  • Limitation of the study and applications should be explained.

Round 2

Reviewer 2 Report

The language should be improved. 

Author Response

Dear Editors and Reviewers:

We would like to thank you for reviewing our manuscript thoroughly and for providing many thoughtful comments.

These reviews help us improve our research and provide the most important guidance for future research.

We've processed the comments to the best of our ability and added some necessary explanations.

To facilitate finding the revised sections, these sections are color-coded in the revised manuscript. Our responses are listed point by point below.

Hope to hear from you soon.

Sincerely,

Sun Yi
